# Knowledge, attitude, and practice of preconception care and associated factors among obstetric care providers working in public health facilities of West Shoa Zone, Ethiopia: A cross-sectional study

Hawi Abayneh[1]*, Negash Wakgari[2], Gemechu Ganfure[3], Gizachew Abdissa Bulto[2]

1 Departmentof Midwifery, College of Medicine and Health Sciences, Salale University, Fitche, Ethiopia,
2 Department of Midwifery, College of Medicine and Health Sciences, Ambo University, Ambo, Ethiopia,
3 Department of Pediatrics and Child Health, College of Medicine and Health Sciences, Ambo University, Ambo, Ethiopia

* negashwakgari@yahoo.com

**Data Availability Statement:** All relevant data are within the paper and its Supporting Information files.

## Abstract

Preconception care is biomedical, social, and behavioural care provided for a woman or couple before conception occurs or throughout their reproductive year. In Ethiopia, it's reported that the majority of health care providers had poor knowledge and practice of pre-conception care. The institution-based cross-sectional study was conducted among 359 obstetric care providers to assess knowledge, attitude, and practice of preconception care in West Shoa Zone, Ethiopia. A stratified, simple random sampling technique selected five hospitals, 46 health centers, and study participants. Pretested and structured question-naires were used to collect data. Data were entered into Epidata and exported to SPSS for analysis. Bivariate and multivariate logistic regressions were employed to identify an associ-ation between the independent predictors and the outcome variables. In this study, 173 (48.2%) and 124(34.5%) of the obstetric care providers had good knowledge and practice of preconception care, respectively. Two-thirds 255(71%) of providers had a favorable attitude toward preconception care. The odds of having good knowledge were higher among Mid-wives' providers [AOR: 2.03, 95%CI: 1.09–3.77] and had training on HIV testing [AOR: 3.5, 95%CI: 1.9–6.4]. The presence of a library [AOR: 1.7, 95%CI: 1.04–2.85] and internet access [AOR: 3.4, 95%CI: 2.0–5.8] in working health facility had a higher odds of good knowledge about preconception. Degree and above holders [AOR: 3.1, 95%CI: 1.5–6.1] also had higher odds of good preconception knowledge than diploma holders. Similarly, the odds of having good practice of preconception care were higher among health care provid-ers: who did screening for reproductive life plans [AOR: 3.7, 95%CI:1.8–7.4], worked in maternity and child health unit [AOR:4.2,95%CI:2.0–8.6], perceive all health facilities should give preconception care services [AOR:2.3,95%CI:1.2–4.3], and perceive all health care providers should provide preconception services [AOR:3.0, 95%CI: 1.7–5.5]. This study found that more than half of obstetric care providers' had poor knowledge, favorable

**Funding:** We are grateful to Ambo University, College of Medicine and Health Science for their financial support. The funders had no role in study design, data collection and analysis, decision to publish, or preparation of the manuscript.

**Competing interests:** The authors have declared that no competing interests exist.

**Abbreviations:** APO, Adverse Pregnancy Outcomes; PCC, Preconception Care; OBCPs, Obstetric Care Providers; HCP, Health Care Provider; WHO, World Health Organization.

attitude, and poor practice of preconception care. Provision of training, carrier development, and installation of internet and library services should be enhanced.

## Introduction

World Health Organization (WHO) defines Preconception care as "biomedical, social, and behavioural care provided for a woman or couples before conception occurs or throughout their reproductive year." To bring a successful preconception care initiative, in 2013, the world health organization (WHO) developed preconception care packages that encompass 13 areas to be addressed by the health care providers [1]. Obstetricians and other obstetric care providers should engage each patient in supportive, respectful conversation about the women's pregnancy intentions and provide pre-pregnancy or contraceptive counseling based on the women's desires and preferences [2].

Study shows that in 2017 global maternal mortality ratio was 211 per 100,000 live births [3], and other evidence indicated that in 2019 the neonatal mortality rate was 17 per 1000 live births [4]. In Ethiopia, the maternal mortality ratio was 412 maternal deaths per 100,000 live births, while under-five children's, infants, and neonatal mortality rates were 67, 45 and 29 per 1000 live births, respectively [5]. Reducing maternal mortality depends on ensuring that ladies have access to quality care before, during, and after childbirth [6]. However, in low and middle-income countries, many women do not have adequate access to the prenatal care they need [7].

The likelihood of using preconception care (PCC) services is dependent on health care workers' knowledge, attitudes, and providing information on preconception care (PCC) to their patients [8]. However, a study done in West Shoa Zone, Oromia, showed that 8.1% of women got information and counseling from the health care providers (HCPs) and 14.5% of them utilize the service [9]. This finding suggests that healthcare providers (HCPs) may have a crucial influence on couples' use of PCC. HCPs have an obligation and take the top place to update evidence-based clinical practices associated with preconception care and have updated knowledge to give PCC [10].

In Ethiopia, some studies show that more than half of healthcare providers have poor knowledge [11–13], and more than 85% of healthcare providers are practicing poorly [14]. Moreover, previous studies have identified different factors for health care providers' knowledge and practice on preconception care, including socio-demographic characteristics, training, attitude towards PCC, availability of policy or protocol on PCC, availability of internet access and library in the working facility [11–14]. Even though some studies were conducted in some parts of Ethiopia on the HCPs' knowledge and practice, the studies were conducted on total health care providers rather than obstetric care providers (OBCPs) [11–15]. There is a lack of information on knowledge, attitude, and practice of preconception care among obstetric care providers in West Shoa Zone, Oromia. As a result, this study intended to assess the Knowledge, attitude, and practice of preconception care and associated factors among obstetric care providers in public health facilities of West Shoa Zone, Oromia, Ethiopia.

## Methods and materials

### Study design, setting and population

An institution-based cross-sectional study was conducted in West Shoa Zone, Oromia regional state, from August 1 to September 8, 2021. West Shoa Zone is one of the zones in the Oromia

Region of Ethiopia. Its administrative city is Ambo town, located 114 kilometers away from the capital city of Ethiopia, Addis Ababa. The West Shoa Zone has 9 Public Hospitals, 92 health centers, and 529 health posts. About 1,085 obstetric care providers were working in maternity and reproductive care units of the Zone. Among those 11 were Obstetricians and Gynecologists, 21 Integrated Emergency Surgical Officers (IESO), 38 General Practitioners, 426 Midwives, 467 nurses, and 222 Health Officers. About 268 OBCPs worked in hospitals during the data collection period, while 817 worked in Health Centers.

All obstetric care providers working in the public health facilities of West Shoa Zone during the study period were considered the source population. In contrast, all obstetric care providers working in West Shoa Zone's selected public health facilities were considered the study population. In addition to this, all obstetric care providers working in public health facilities of West Shoa Zone during the data collection period were included in the study, and those OBCPs who served for less than six months were excluded from the study.

## Sample size determination and sampling procedure

The sample size for the first two specific objectives: to determine knowledge and preconception care practice among obstetrics care providers, was calculated by a single population proportion formula n = (Z $\alpha$/2)2 P (1-P)/d2 based on the following assumptions: the proportion (P) of knowledge and practice were 31% [11] and 19.2% [15] respectively which was taken from the previous studies, 95% confidence level of Z $\alpha$/2 = 1.96, 5% of absolute precision. Thus, a 10% non-response rate gave 362 and 262, respectively, and the final sample size became 362.

The facilities were stratified into hospitals and health centers. With a simple random sampling technique, 5(N = 157) hospitals and 46(N = 401) health centers were selected. The sample was allocated to each stratum proportionally based on the number of health care providers working at the selected hospitals and health centers, 157/558*362 = 102 OBCPs from hospitals and 401/558*362 = 260 OBCPs from health centers. Then, a simple random sampling technique was used to select study participants.

## Data collection tools and procedures

The data were collected using structured self-administered questionnaires. The questionnaire was adapted from a previous study conducted in Ethiopia [11] and had five sections (sociodemographic information, knowledge, attitude and practice, and associated factors of preconception care questions). The knowledge section had 15 knowledge-related questions. A 1 and 0 were given for correct and incorrect answers, respectively. Then, HCPs who scored $\geq$ 50th percentile were considered good knowledge of PCC.

The practice section had 34 questions with three PCC practice components measuring the frequency of practice in the last six months of the period. Each question have an option response of never = 0, rarely = 1, sometimes = 2, often = 3, and always = 4. This gives a minimum score of 34*0 = 0 to the maximum score of 34*4 = 136. In this study, those OBCPs who scored $\geq$ 50% were considered good PCC practice; otherwise poor.

Similarly, the OBCPs' attitudes were assessed by their level of agreement on nine questions using the Likert scale. Those OBCPs who scored 60% and above the possible maximum score were considered OBCPs with a favorable attitude towards PCC; otherwise, they were considered OBCPs with an unfavorable attitude toward PCC. The questionnaires were disseminated to the OBCPs and facilitated by six trained graduated unemployed nurses and supervised by two senior BSc midwives.

## Data quality assurance

To assure quality of the study, data collectors and supervisors were given two days of training about the study materials and data collection procedures. Before actual data collection, the study tool was pre-tested among 18 OBCPs working at Tulu Bolo hospital in South West Shoa Zone, Ethiopia. The tool's reliability was checked for its internal consistency with a Cronbach's α test, 0.802 and 0.97 for knowledge and practice, respectively. Moreover, the completeness and consistency of the collected data were reviewed and checked by supervisors and investigators.

**Operational definition.** *Obstetric care providers*. Certified obstetricians and gynecologists, general practitioners, integrated emergency surgical officers, nurses, midwives, and public health officers working in maternal and reproductive health care units [16].

*Knowledge of PCC*. Respondents who scored less than the 50th percentile of the knowledge-related items were categorized as HCPs with 'poor PCC knowledge.' In contrast, HCPs who scored ≥ 50th percentile were considered good knowledge of PCC [12].

*Attitude towards PCC*. Attitude was measured using nine questions with possible five-point Likert scale responses. Those OBCPs who scored 60% and above the possible maximum score [5*9 = 45] were considered OBCPs with a favorable attitude towards PCC; otherwise, they were considered OBCPs with unfavorable attitudes towards PCC.

*OBCP's PCC practice*. OBCPs who scored < 50% of PCC practice items were classified as practitioners demonstrating poor PCC practice. Those OBCPs who scored ≥ 50% were considered good PCC practice [14].

## Data analysis procedures

The collected data were cleaned, coded, and entered into the Epidata version 3.1 and exported to SPSS version 20. Coding was reversed in negative statements. The frequency, proportion, mean, and standard deviation were performed for dependent and independent variables.

Binary logistic regression was done to identify candidate variables for multiple logistic regressions. Then those candidate variables were analyzed with multivariate logistic regression, and those variables at a p-value of < 0.05 were considered to have a statistically significant association with the outcome variables. Odds ratios with 95%CI were used to test the strength of association. Multicollinearity assumption was checked by the variance inflation factor (VIF) of < 1.2, which indicated a less likely correlation between independent variables. The models' goodness of fit was tested using the Hosmer-Lemeshow test, and it was a good fit.

## Ethics statement

The ethical approval was obtained from Ambo University ethical review board. Written Informed consent was obtained from each participant. The participants were assured that participation in the study was voluntary and that they could withdraw during the study. The collected raw data was kept confidential in a secure place, and the names of the participants were not written in the study record. Participants' rights to anonymity and confidentiality were fully protected. All of the information given by participants was recorded in a manner that did not link the respondents with the data.

## Results

### Socio-demographic characteristics of the respondents

Out of the total samples, 359 obstetric care providers participated with a 99% response rate. More than half, 183(51%) of the participants were male. Of all participants, 145(40.4%)

werefound between 26–30 years old. More than half, 207(57.7%) of respondents had more than five years of service experience, and 285(79.4%) had degreesand above (Table 1).

## Obstetric care provider's knowledge about preconception care

Out of the total participants, 173(48.2%) [95%CI: 43.2–53.5] of OBCPs had a good knowledge of preconception care. The knowledge score range from 1to 12. The majority, 233(64.9%), knew that the eligible clients for PCC include all adolescents and reproductive age individuals (Table 2).

## Attitude toward preconception care

Two-thirds, 255(71%) of OBCPS had a favorable attitude towards PCC. Out of the total 45 scores, respondents scored a minimum of 12 and a maximum of 41, with a mean score of 28.1 with SD ± 4.96. More than half, 216(60.2%) of the respondents strongly disagreed that providing PCC is not within the scope of my professional responsibility and accountability. About one-third, 127(35.5%) of respondents strongly agreed with the idea that omission of preconception care leads to irreversible damage to the fetus (Table 3).

## Preconception care practice among health care providers

Out of the total participants, 124(34.5%) of the providers had a good practice of PCC.The Obstetric care providers' practice scores range from 1 to 136 out of the possible maximum 136 points score. Despite their differences, all participants have practised at least a single component of the PCC practice question.

**Table 1. Obstetric care providers' socio-demographic characteristics in West Shoa Zone, Oromia, Ethiopia, 2021(n = 359).**

| Variables | | Frequency | Percentage |
|---|---|---|---|
| Gender | Male | 183 | 51.0 |
| | Female | 176 | 49.0 |
| Age group in years | 20–25 | 17 | 4.7 |
| | 26–30 | 145 | 40.4 |
| | 31–35 | 97 | 27.0 |
| | ≥36 | 100 | 27.9 |
| Marital status | Single | 119 | 33.1 |
| | Married | 188 | 52.4 |
| | Divorced | 30 | 8.4 |
| | Widowed | 8 | 2.2 |
| | Living together | 14 | 3.9 |
| Profession | Midwives | 168 | 46.8 |
| | Nurses | 88 | 24.5 |
| | General practitioners | 27 | 7.5 |
| | Public health officers | 66 | 18.4 |
| | Others* | 10 | 2.8 |
| Working institution | Hospital | 102 | 28.4 |
| | Health center | 257 | 71.6 |
| Working experience inyears | ≤ 5 | 152 | 42.3 |
| | >5 | 207 | 57.7 |
| Educational level | Diploma | 74 | 20.6 |
| | Degree and above | 285 | 79.4 |
| Working unit | MCH | 108 | 30.1 |
| | Gynecologic OPD | 67 | 18.7 |
| | Gynecologic ward | 51 | 14.2 |
| | Labor and deliveryward | 133 | 37.0 |

**Table 2. Obstetric care providers' knowledge about preconception care in West Shoa Zone, Oromia, Ethiopia, 2021(n = 359).**

| Variables | Response category | Frequency | Percentage |
|---|---|---|---|
| The eligible clients for PCC include all adolescents and reproductive age individuals | Yes | 233 | 64.9 |
| | No | 122 | 34.0 |
| | Do not know | 4 | 1.1 |
| To be effective, PCC should start four weeks before conception | Yes | 190 | 52.9 |
| | No | 162 | 45.1 |
| | Do not know | 7 | 1.9 |
| Periodontal disease is a risk factor for adverse pregnancy outcomes (APO) | Yes | 57 | 15.9 |
| | No | 129 | 35.9 |
| | Do not know | 173 | 48.2 |
| Planning pregnancy with a BMI of ≤ 18.4 increases the risk of developing APO | Yes | 286 | 79.7 |
| | No | 43 | 12.0 |
| | Don't know | 30 | 8.4 |
| All women of reproductive age should take 400 mcg of folic acid daily | Yes | 255 | 71.0 |
| | No | 83 | 23.1 |
| | Do not know | 21 | 5.8 |
| Women need to start taking folic acid 3months before pregnancy | Yes | 77 | 21.4 |
| | No | 211 | 58.8 |
| | Do not know | 71 | 19.8 |
| The recommended routine preconception laboratory tests include Hct, HIV,HBV, and RPR or VDRL tests | Yes | 79 | 22.0 |
| | No | 208 | 57.9 |
| | Do not know | 72 | 20.1 |
| Preconception genetic counseling and screening include recommending carrier screening tests for the client with sickle cell hemoglobinopathies | Yes | 93 | 25.9 |
| | No | 234 | 65.2 |
| | Do not know | 32 | 8.9 |
| Isotretinoin, Valproic acid, and Warfarin are medications that pose teratogenic effects requiring preconception modification | Yes | 120 | 33.4 |
| | No | 150 | 41.8 |
| | Do not know | 89 | 24.8 |
| Early identification and treatment of diseases like depression, seizure disorder, and phenylketonuria during the preconception period reduce the occurrence of APO | Yes | 117 | 32.6 |
| | No | 179 | 49.9 |
| | Do not know | 63 | 17.5 |
| The recommended test that guarantees good preconception blood sugar control for a woman with pregestational diabetes is the random blood sugar test | Yes | 121 | 33.7 |
| | No | 217 | 60.4 |
| | Do not know | 21 | 5.8 |
| Recommending regular exercise is an important PCC counseling point. Thus, women planning pregnancy should aim for 30 minutes of moderate exercise 5 days a week | Yes | 110 | 30.6 |
| | No | 183 | 51.0 |
| | Do not know | 66 | 18.4 |
| Women planning pregnancy should be advised to delay pregnancy until reducing drug, alcohol, and tobacco use | Yes | 218 | 60.7 |
| | No | 129 | 35.9 |
| | Do not know | 12 | 3.3 |
| A clinician attending to clients with previous caesarian section should advise the client to delay the next pregnancy for at least 18 months before the next conception | Yes | 285 | 79.4 |
| | No | 66 | 18.4 |
| | Do not know | 8 | 2.2 |
| Infertility screening and management is not the concern of PCC | Yes | 243 | 67.7 |
| | No | 67 | 18.7 |
| | Do not know | 49 | 13.6 |

**Table 3. Obstetric care providers' attitude towards preconception care in West Shoa Zone, Oromia, Ethiopia, 2021(n = 359).**

| Variables | Response category | Frequency | Percentage |
|---|---|---|---|
| Omission of preconception care leads to an irreversibledamage to the fetus | Strongly disagree | 125 | 34.8 |
| | Disagree | 36 | 10 |
| | Undecided | 26 | 7.2 |
| | Agree | 45 | 12.5 |
| | Strongly agree | 127 | 35.5 |
| PCC provides the most incredible opportunity to optimize couples' health particularly women's health before conception | Strongly disagree | 120 | 33.4 |
| | Disagree | 32 | 8.9 |
| | Undecided | 21 | 5.8 |
| | Agree | 45 | 12.5 |
| | Strongly agree | 141 | 39.3 |
| Providing PCC services to developing countries like Ethiopia is a luxury service | Strongly disagree | 90 | 25.1 |
| | Disagree | 139 | 38.7 |
| | Undecided | 64 | 17.8 |
| | Agree | 42 | 11.7 |
| | Strongly agree | 24 | 6.7 |
| In developing countries like Ethiopia, the focus of PCC should not be directed to healthy people but to people with infectious diseases like HIV and HBV | Strongly disagree | 201 | 56 |
| | Disagree | 62 | 17.3 |
| | Undecided | 34 | 9.5 |
| | Agree | 42 | 11.7 |
| | Strongly agree | 20 | 5.6 |
| Providing PCC is not within the scope of my professional responsibility and accountability | Strongly disagree | 216 | 60.2 |
| | Disagree | 69 | 19.2 |
| | Undecided | 33 | 9.2 |
| | Agree | 31 | 8.6 |
| | Strongly agree | 10 | 2.8 |
| Due to the presence of other competing demands, providing PCC is not the priority intervention I should provide | Strongly disagree | 203 | 56.5 |
| | Disagree | 72 | 20.1 |
| | Undecided | 39 | 10.9 |
| | Agree | 32 | 8.9 |
| | Strongly agree | 13 | 3.6 |
| Preconception care should be given for all healthy and sick individuals including those presented with a critical and emergency condition | Strongly disagree | 224 | 62.4 |
| | Disagree | 75 | 20.9 |
| | Undecided | 31 | 8.6 |
| | Agree | 24 | 6.7 |
| | Strongly agree | 5 | 1.4 |
| All healthcare providers can easily integrate the elements of PCC in their daily practice to all eligible individuals whom they are caring | Strongly disagree | 144 | 40.1 |
| | Disagree | 70 | 19.5 |
| | Undecided | 60 | 16.7 |
| | Agree | 24 | 6.7 |
| | Strongly agree | 61 | 17 |
| Preconception health is part of the reproductive andthe human rights issue to which the health professional is responsible either for omission or commission of PCC | Strongly disagree | 186 | 51.8 |
| | Disagree | 108 | 30.1 |
| | Undecided | 23 | 6.4 |
| | Agree | 27 | 7.5 |
| | Strongly agree | 15 | 4.2 |

## Factors associated with the knowledge of obstetric care providers on preconception care

In this study, Midwives were two times more likely to have a good knowledge than nurses [AOR: 2.03, 95%CI: 1.09–3.77]. The likelihood of having good knowledge of PCC was three-fold higher among degree and above holders than those diploma holders [AOR: 3.11, 95%CI: 1.57–6.15]. Those OBCPs who read PCC guidelines or protocols from any source were two times more knowledgeable than those who never read PCC guidelines or protocols which are developed by other countries [AOR: 1.85, 95%CI: 1.09–3.12].OBCPs who were working in hospitals were two times more knowledgeable than those professionals who were working in health centres [AOR: 2.12, 95%CI: 1.1–3.8]. The other associated factor found in this study was, having training on HIV testing and management. Those OBCPs who had training on HIV testing and management were more than three-folds knowledgeable as those who did not have the training [AOR 3.51, 95%CI: 1.93–6.40]. OBCPs who work in a facility that had a library were two times more knowledgeable than those whose facilities had no library [AOR: 1.73, 95%CI: 1.04–2.85]. Those professionals who were working in facilities that had internet access were more than three folds knowledgeable as those whose facilities had no internet access [AOR: 3.45, 95%CI: 2.05–5.81] (Table 4).

**Table 4. Binary and multivariate logistic regression analysis results of preconception care knowledge among obstetric care providers, West Shoa Zone, Oromia, Ethiopia, 2021.**

| Variables | | OBCPs knowledge on PCC | | COR(95%C.I) | AOR(95% C.I) | P-value |
|---|---|---|---|---|---|---|
| | | Good | Poor | | | |
| Gender | Male | 95(51.9%) | 88(48.1%) | 1.35(0.89–2.05) | 1.28(0.77–2.12) | 0.33 |
| | Female | 78(44.3%) | 98(55.7%) | 1* | 1* | |
| Profession | Doctor | 23(67.6%) | 11(32.4%) | 3.84(1.6 5–8.91) | 1.30(0.44–3.82) | 0.62 |
| | Nurse | 31(35.2%) | 57(64.8%) | 1* | 1* | 0.02** |
| | Midwife | 95(56.5%) | 73(43.5%) | 2.39(1.40–4.07) | 2.03 (1.09–3.77) | 0.06 |
| | Health officer | 24(34.8%) | 45(65.2%) | 0.98(0.50–1.89) | 0.4 7(0.21–1.05) | |
| Educational level | Degree and above | 147(51.6%) | 138(48.4%) | 1.96(1.15–3.34) | 3.11(1.57–6.15) | 0.001** |
| | Diploma | 26(35.1%) | 48(64.9%) | 1* | 1* | |
| Working institution | Hospital | 67(65.7%) | 35(34.3%) | 2.72(1.6 9–4.39) | 2.12(1.17–3.84) | 0.01** |
| | Health center | 106(41.2%) | 151(58.8%) | 1* | 1* | |
| Ever read PCC guidelines from any source | Yes | 120(57.4%) | 89(42.6%) | 2.46(1.60–3.80) | 1.85(1.09–3.12) | 0.02** |
| | No | 53(35.3%) | 97(64.7%) | 1* | 1* | |
| Training on HIV testing and management | Yes | 143(57%) | 108(43%) | 3.44(2.11–5.61) | 3.51 (1.93–6.40) | 0.00** |
| | No | 30(27.8%) | 78(72.2%) | 1* | 1* | |
| Training on providing alcohol or tobacco cessation service | Yes | 130(53.5%) | 113(46.5%) | 1.95(1.24–3.07) | 1.04(0.5–1.90) | 0.89 |
| | No | 43(37.1%) | 73(62.9%) | 1* | 1* | |
| Presence of library in working health facility | Yes | 104(56.2%) | 81(43.8%) | 1.95(1.28–2.97) | 1.73(1.04–2.85) | 0.03** |
| | No | 69(39.7%) | 105(60.3%) | 1* | 1* | |
| Presence of internet access in working health facility | Yes | 115(59.6%) | 78(40.4%) | 2.74(1.78–4.21) | 3.45(2.05–5.81) | 0.00** |
| | No | 58(34.9%) | 108(65.1%) | 1* | 1* | |
| The perceived expectation on who should give PCC | All health care providers | 72(53.3%) | 63(46.7%) | 1.39(0.90–2.13) | 1.48(0.87–2.54) | 0.14 |
| | Selected health care provider | 101(45.1%) | 123(54.9%) | 1* | 1* | |
| Opinion on which health facility should give PCC services | All health facility | 61(58.1%) | 44(41.9%) | 1.75(1.11–2.78) | 1.35(0.76–2.41) | 0.29 |
| | Selected health facility | 112(44.1%) | 142(55.9%) | 1* | 1* | |

*Reference,

** Significant at P-Value <0.05, COR- Crude Odds Ratio, AOR-Adjusted Odds Ratio, CI = Confidence Interval.

## Factors associated with obstetric care providers' practice on preconception care

Participants' knowledge of preconception care had no significant association with their level of practice. Obstetric care providers who were working in MCH [AOR: 3.90, 95%CI: 1.94–7.82] and Gynecologic OPD [AOR: 3.74, 95%CI: 1.69–8.24] had a more likelihood of PPC practice compared to OBCPs who was working in the labour& delivery ward. The likelihood of good practice of PCC was four-fold higher among obstetric care providers who did screening for RPL plan of clients than those who didn't screen [AOR: 3.51, 95%CI: 1.76–6.98]. In addition, those professionals who ever read preconception guidelines and protocol from any source were two times more likely to have good practice of PPC than those who never read [AOR: 1.82, 95% CI: 1.03–3.23].

OBCPs who was working in a health facility that had internet access were three times more likely to have good practice than those OBCPs who were working in a health facility that did nothave an internet access [AOR:3.29, 95% CI: 1.82–5.95].OBCPs who possessed a perceived expectation that PCC should be given by all health care professionals had higher odds of PCC practice than those who perceived PCC should be given by selected professionals [AOR: 3.02, 95% CI:1.69–5.41)]. The odds of good PCC practice was two times higher among OBCPs who perceived PCC should be given in all health facility than those who perceived PCC should be given in selected health facility [AOR: 2.37,95% CI:1.29–4.33]. The likelihood of good practice of PCC was two times higher among obstetric care providers who had training on RPL plan screening and counselling than those who did nothave [AOR:1.77,95%CI: 1.01–3.10] (Table 5).

## Discussion

This study revealed that 48.2% [95%CI: 43.2–53.5] of OBCPs had a good knowledge of PCC. This finding is consistent with the studiesdone in Brazil (46.2%) [17], Nigeria 52.7% [18], Awi, Ethiopia 52% [12], and Wollo, Ethiopia(49.1%) [13]. However, it is higher than a study done in Hawassa(31%) [11]. This difference may be due to the concept of PCC was a new initiative at the time of the study, and the non-inclusion of PCC courses in pre-service training [19], and currently, the care was included in pre-service curricula of some programs. The present finding is lower than the studies done in TikureAnbessa Hospital Ethiopia 69.2% [15], Iran (88.3%) [20], Nepal (85.9%) [21], and South Africa (55%) [22]. This might be due to, unlike in Ethiopia PCC service existed and was implemented earlier, and also it was considered to be a part of integrated care and responsibility of providers so that this can increase the provider's exposure to PCC in the case of Iran [23]. The possible reason for the study of Nepal and South Africa might be due to differences in sample size and sampling technique. The other possible explanation for the differences with the study of TikurAnbessa might be due to the study participants' academic profile, being the largest teaching and referral hospital in Ethiopia and the current study was done in urban and more rural areas which intern affects the provider's accessibility to information and updated knowledge.

This study revealed that 34.5% (95%CI: 29.5–39.6) of OBCPs had a good practice on PCC. This finding was lower than the studies done in Canada (78.4%) [24] and Netherlands 82% [25]. This difference may be due to PCC service being a new initiative that is yet to be introduced in the health care system of Ethiopia and in those countries dedicated PCC service unit was established for the provision of PCC [10]. The other reason might be, unlike those countries, Ethiopia did notyet develop PCC practice guidelines rather preconception guidance was usually contained within maternity guidelines [19]. However, the current finding was higher than the study conducted at TikurAnbessa hospital where is 19.2% of residents had a good level of preconception care practice [15]. This difference may be due to the samplesize difference.

**Table 5. Binary and multivariate logistic regression analysis results of preconception care practice among obstetric care providers, West Shoa Zone, Oromia, Ethiopia,2021.**

| Factors | | OBCPs practice on PCC | | COR (95.0%C.I) | AOR(95.0% C.I) | P-value |
|---|---|---|---|---|---|---|
| | | Good | Poor | | | |
| Age | 20–25 | 8(47.1%) | 9(52.9%) | 1* | 1* | |
| | 26–30 | 57(39.3%) | 88(60.7%) | 0.72(0.26–1.99) | 1.20(0.36–3.94) | 0.76 |
| | 31–35 | 32(33%) | 65(67%) | 0.55(0.19–1.57) | 0.86(0.24–3.11) | 0.82 |
| | ≥36 | 27(27%) | 73(73%) | 0.41(0.14–1.18) | 0.76(0.20–2.95) | 0.70 |
| Working experience | >5 years | 63(30.4%) | 144(69.6%) | 0.65(0.42–1.01) | 0.56(0.29–1.07) | 0.07 |
| | ≤5 years | 61(40.1%) | 91(59.9%) | 1* | 1* | |
| Working unit | Maternal child health care | 55(50.9%) | 53(49.1%) | 5.23(2.89–9.47) | 3.90(1.94–7.82) | 0.00** |
| | Gynecologic OPD | 28(41.8%) | 39(58.2%) | 3.62(1.85–7.05) | 3.74(1.69–8.24) | 0.001** |
| | Gynecologic ward | 19(37.3%) | 32(62.7%) | 2.99(1.44–6.21) | 2.23(0.96–5.18) | 0.06 |
| | Labor anddelivery ward | 22(16.5%) | 111(83.5%) | 1* | 1* | |
| Attitude | Favorable | 93(36.5%) | 162(63.5%) | 1.35(0.82–2.21) | 1.26(0.66–2.38) | 0.47 |
| | Unfavorable | 31(29.8%) | 73(70.2%) | 1* | 1* | |
| RPL plan screening | Screening | 107(42.1%) | 147(57.9%) | 3.76(2.11–6.70) | 3.51(1.76–6.98) | 0.00** |
| | Not screening | 17(16.2%) | 88(83.8%) | 1* | 1* | |
| Ever read PCC guideline or protocol from any source | Yes | 87(41.6%) | 122(58.4%) | 2.17(1.37–3.45) | 1.82(1.03–3.23) | 0.03** |
| | No | 37(24.7%) | 113(75.3%) | 1* | 1* | |
| Training on PCC consideration for clients with chronicdisease | Yes | 88(40.4%) | 130(59.6%) | 1.97(1.24–3.14) | 1.70(0.96–3.01) | 0.06 |
| | No | 36(25.5%) | 105(74.5%) | 1* | 1* | |
| Training on RPL plan screening and counseling | Yes | 56(39.2%) | 87(60.8%) | 1.40(0.90–2.17) | 1.77(1.01–3.10) | 0.04** |
| | No | 68(31.5%) | 148(68.5%) | 1* | 1* | |
| Presence of internet access in working health facility | Yes | 88(45.6%) | 105(54.4%) | 3.02(1.90–4.82) | 3.29(1.82–5.95) | 0.00** |
| | No | 36(21.7%) | 130(78.3%) | 1* | 1* | |
| Perceived expectation on who should give PCC | All health care professionals | 67(49.6%) | 68(50.4%) | 2.88(1.83–4.53) | 3.02(1.69–5.41) | 0.00** |
| | The selected health care professional | 57(25.4%) | 167(74.6%) | 1* | 1* | |
| Opinion on which health facility should give PCC services | All health facility | 55(52.4%) | 50(47.6%) | 2.94(1.8 3–4.73) | 2.37(1.2 9–4.33) | 0.005** |
| | Selected health facility | 69(27.2%) | 185(72.8%) | 1* | 1* | |

*Reference,

** Significant at P-Value <0.05,COR- Crude Odds Ratio, AOR-Adjusted Odds Ratio, CI = Confidence Interval.

In this study, participant's profession was associated with obstetric care providers' knowledge about PCC. Midwives were two times more likely to have agood knowledge of PCCcompared to nurses. This finding is in line with the study conducted in Wollo, Ethiopia [13]. The knowledge gap might be due to, unlike nurses, midwives might have chances to be exposed to preconception care during their pre-service training [26] and their working departments being linked to the components of PCC.

The likelihood of having good knowledge of PCC was three-fold higher among degree andabove holders compared to diploma holders. This finding was supported by a study conducted in Nepal [21]. This might be due to the educational curriculum differences.

Those OBCPs who read PCC guidelines or protocol from any source were two times more knowledgeable than those who never read PCC guidelines or protocol. This finding was supported by the studies done in Wollo, Ethiopia [13] and Hawassa, Ethiopia [11]. This finding is considerable in that getting access to different guidelines and procedural documents that guide PCC will enable providers to be informed and to know the components of PCC services.

OBCPs who were working in the hospitals were two times more knowledgeable than those professionals who were working in the health centre. This finding is also in line with a study done in Hawassa [11]. This might be due to the presence of different higher specialized care; they had a chance of exposure to different clinical cases than health centres.

The other factor found in this study is having training on HIV testing and management. Those OBCPs who had training on HIV testing and management were more than three-folds knowledgeable than those who did nothave the training. The finding was in line with the studies conducted in Nepal [21] and Awi, Ethiopia [12]. This is literal in that when professionals' get in-service training they will get updated knowledge and the HIV training manuals included preconception care components [27].

OBCPs who work in an institution that had library and internet access were two times and more than three-fold knowledgeable respectively. This finding was supported by a study done in Wollo, Ethiopia [13]. This is literal in that, HCPs can access those guidelines, different medical-related books, and manuals for reading, gathering, and updating information therefore this can intern will increase their knowledge [28,29].

The likelihood of good practice of PCC was fourfold higher among obstetric care providers who did screening for RPL plan of clients than those who did notscreen; this finding was in line with a study done in Hawassa, Ethiopia [14]. This is possible in that requesting women's RPL plan could serve as an opening wedge to initiate several evidence-based pre-pregnancy care interventions [30].

Those professionals who ever read preconception guidelines and protocol from any source were two times more likely to have agood practice than those who never read. This finding is also supported by studies done in the Netherlands and Australia [31–33]. This is possible in that knowing guidelines about PCC will serve as guidance and motivate to give effective PCC services.

OBCPs that possessed a perceived expectation that PCC should be given by all health care professionals had odds of good PCC practice by three-fold than those who perceived PCC should be given by selected professionals. This finding was in line with a study done in Hawassa [14]. This is possible in that having this opinion makes the professional himself take responsibility to give the care.

## Strength and limitations of the study

This study considers all health care providers responsible for the provision of preconception care. However, it has some limitations: Firstly, this study was not supported with observational studies for the assessment of preconception care practice. Hence, we cannot validate their actual performance. Secondly, there might be the possibilities of recall and social desirability bias.

## Conclusions

This study found that more than half of obstetric care providers' had poor knowledge and practice of preconception care. Respondents' profession, educational level, type of working institution, ever read PCC guidelines from any source, having training on HIV testing and management and institutions having library andinternet access were associated with the providers' knowledge. On the other hand, participants working unit, training on RPL plan screening andcounselling, RPL plan screening, ever read PCC guidelines or protocol from any source, presence of internet access in working health facilities, perceived expectations on who should give PCC, and opinion on which health facility should give PCC services were

associated to the providers' good practice. Provision of training, carrier development, and installation of internet and library services in working facilities is recommended.

## Supporting information

**S1 Data set.**
(SAV)

**S1 Questionnaire.**
(PDF)

## Acknowledgments

We are grateful to Ambo University, College of Medicine, and Health sciences staff for their guidance and support during the data collection process. Finally, we would like to appreciate the West Shoa zone health office and respondents

## Author Contributions

**Conceptualization:** Hawi Abayneh, Negash Wakgari, Gemechu Ganfure, Gizachew Abdissa Bulto.

**Data curation:** Hawi Abayneh, Negash Wakgari, Gemechu Ganfure, Gizachew Abdissa Bulto.

**Formal analysis:** Hawi Abayneh, Negash Wakgari, Gizachew Abdissa Bulto.

**Funding acquisition:** Hawi Abayneh, Negash Wakgari, Gemechu Ganfure.

**Investigation:** Hawi Abayneh, Gemechu Ganfure, Gizachew Abdissa Bulto.

**Methodology:** Hawi Abayneh, Negash Wakgari, Gizachew Abdissa Bulto.

**Project administration:** Hawi Abayneh, Negash Wakgari, Gemechu Ganfure, Gizachew Abdissa Bulto.

**Resources:** Hawi Abayneh, Negash Wakgari, Gemechu Ganfure, Gizachew Abdissa Bulto.

**Software:** Hawi Abayneh, Negash Wakgari, Gemechu Ganfure.

**Supervision:** Hawi Abayneh, Negash Wakgari, Gemechu Ganfure, Gizachew Abdissa Bulto.

**Validation:** Hawi Abayneh, Negash Wakgari, Gizachew Abdissa Bulto.

**Visualization:** Hawi Abayneh, Negash Wakgari, Gemechu Ganfure.

**Writing – original draft:** Hawi Abayneh, Negash Wakgari, Gemechu Ganfure.

**Writing – review & editing:** Hawi Abayneh, Negash Wakgari, Gizachew Abdissa Bulto.

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
