## [Decision Letter · Decision Letter 0]

11 May 2022

PONE-D-22-10177Knowledge and practice of preconception care and associated factors among obstetric care providers working in public health facilities of West Shoa Zone, Ethiopia: A cross-sectional studyPLOS ONE

Dear Dr. Wakgari,

Thank you for submitting your manuscript to PLOS ONE. After careful consideration, we feel that it has merit but does not fully meet PLOS ONE’s publication criteria as it currently stands. Therefore, we invite you to submit a revised version of the manuscript that addresses the points raised during the review process.

We look forward to receiving your revised manuscript.

Kind regards,

Jayanta Kumar Bora

Academic Editor

PLOS ONE

Journal Requirements:

Additional Editor Comments:

Dear Mr. Negash Wakgari,

Thank you for submitting your manuscript “Knowledge and practice of preconception care and associated factors among obstetric care providers working in public health facilities of West Shoa Zone, Ethiopia: A cross-sectional study” to PLOS ONE. After careful consideration, we feel that it has merit but does not fully meet PLOS ONE publication criteria as it currently stands. Therefore, we invite you to submit a revised version of the manuscript that addresses the points raised by the reviewers during the review process. Overall you need to strengthen in editing, analysis and interpretation areas. You have to address the reviewer comments in bullet point wise while submitting the revised version of the paper.

We would appreciate receiving your revised manuscript by June 8 2022. When you are ready to submit your revision, log on to https://pone.editorialmanager.com/ and select the 'Submissions Needing Revision' folder to locate your manuscript file.

Dr. Jayanta Bora

Academic Editor

PLOS ONE

Reviewer’s Comment

Reviewer1

General Comment:

In general, it is a well design study, conducted in a way scientifically and technically sound methodology. Appropriate data analysis method was employed, and figures are consistent through the document. Nevertheless, this manuscript needs editorial revision and minor changes to the abstract and discussion to make it clear and more attractive for the reader.

Detail line by line comments are provided below.

Title:

L112-118: the Data collection tools and procedures indicate that the study tool has five sections(demographic information, knowledge, attitude and practice, and associated factors of preconception care questions). The proceeding sentence also shows that you have assess attitude as well. Therefore, attitude should be included in the title, keywords, and result sections.

Since the data collection tool(questionnaire) is not provided, I couldn’t verify the actual questions that address the attitude section. Please, provide the study materials with your revised manuscript.

Abstract

L27-33: Please paraphrase the result section of the abstract.

•L27-30: has too many conjunctions “and/&” and it is unclear.

•Select the most relevant findings to be included in the abstract.

L32: "doing screening for reproductive life plan"

oThis is a very specific task. I suggest using professional category or the department instead of pointing out a very specific task. Also, it would be much better if the result can be presented in carrier-wise like diploma, degree, midwives, physicians, and other gyn & obs care providers in the selected health facilities.

Keywords:

Add attitude if providers’ attitude was assessed, otherwise the attitude assessment questionnaire should be removed from the tool(See comment L112-118).

Methods

L98: “ the first two specific objectives” - Please mention each objective here.

L106-107: Please, paraphrase as “ The sample was allocated to each stratum proportionally based on the number of health care providers working at the selected hospitals and health centers”.

L114 - Indicates that knowledge, attitude, and practice preconception care of participants were assessed. See my comments above.

L122 -124: Data quality assurance is beyond a questionnaire that includes data collection, data processing and data management. So, please paraphrase as

"To assure quality of the study, data collectors and supervisors were given two days training about the study materials and data collection procedures. The study tool was pre-tested among ...... "

L127: “… Cronbach's α tests which were 0.802 & 0.97 for knowledge and practice, respectively.”

L172 - “1 -to 12” correct as 1 to 12

L174 - “ 7-to 136” correct as 1 to 136

L175-176: I don't think this sentence is important. The above sentence showed that how many of them perform good practice based on your classification of good and poor.

Otherwise, you can explain as "only three participants were found to perform all the expected good practice on preconception care.... It is better focus on the meaning than the numbers.

L187: I think the health centers are also public. This can mislead to a comparison of public and non-public sectors.

Discussion and conclusion

oPlease include strengths and limitations of this study.

Reviewer2

Dear authors’ thank you for submitting this nice manuscript.

Here under some comments about your manuscript.

1. You have used special character (&) in the abstract part revise it.

2. You have used stratified sampling technique in your main text part, but also stated as simple random sampling technique in the abstract part revise it.

3. What about the interaction effect between knowledge and practice? Try to put it clearly.

4. What is the advantages of computing two sample size in one study? Is there any scientific study that support your ways of computing sample size? Justify it.

5. What is the measurement scale for your dependent variables? Means that the knowledge and practice is not directly measurable, by what criteria have you categorized it. Put the justification clearly in your document. Look the following statement from your main text

“After correct responses were given a score of 1 and incorrect answers were given a score of 0, the first dependent variable, knowledge score, was computed for each respondent. The practice score, the second dependent variable, was computed for each respondent. 'Never = 0, rarely = 1, sometimes = 2, often = 3, and always = 4' were the response alternatives. “Your dependent variables are dichotomous, how do you categorized them? Justify it.

6. Try to cite any scientific work which supports your statement “Binary logistic regression was done to identify those candidate variables for Multiple logistic regression at a p-value of < 0.25 for not to miss the important variables.”

7. In your result part you have used only descriptive statistics. Why not Cross tab? Try to analyze your data again by using cross tab and interpret the result by considering each categories of dependent variable with each categories of independent variables. For instance: which age group of individuals have better practiced? So in order to answer questions like this you have to use cross tab rather using simple descriptive statistics.

8. Why not you haven’t seen association between the dependent and independent variable, by using Chi-square test?

9. Try to modify table 2 & 3 title, because the result is not only the binary logistic regression.

10. Interpret the cross tab result presented in the table 2 & 3, it is not interpreted.

In general your manuscript needs major revision.

Reviewers' comments:

Reviewer's Responses to Questions

**Comments to the Author**

1. Is the manuscript technically sound, and do the data support the conclusions?

Reviewer #1: Yes

Reviewer #2: Yes

2. Has the statistical analysis been performed appropriately and rigorously? 

Reviewer #1: Yes

Reviewer #2: Yes

3. Have the authors made all data underlying the findings in their manuscript fully available?

Reviewer #1: Yes

Reviewer #2: Yes

4. Is the manuscript presented in an intelligible fashion and written in standard English?

Reviewer #1: No

Reviewer #2: Yes

5. Review Comments to the Author

Reviewer #1: In general, it is a well design study, conducted in a way scientifically and technically sound methodology. Appropriate data analysis method was employed, and figures are consistent through the document. Nevertheless, this manuscript needs editorial revision and minor changes to the abstract and discussion to make it clear and more attractive for the reader.

Reviewer #2: Dear authors’ thank you for submitting this nice manuscript.

Here under some comments about your manuscript.

1.You have used special character (&) in the abstract part revise it.

2.You have used stratified sampling technique in your main text part, but also stated as simple random sampling technique in the abstract part revise it.

3.What about the interaction effect between knowledge and practice? Try to put it clearly.

4.What is the advantages of computing two sample size in one study? Is there any scientific study that support your ways of computing sample size? Justify it.

5.What is the measurement scale for your dependent variables? Means that the knowledge and practice is not directly measurable, by what criteria have you categorized it. Put the justification clearly in your document. Look the following statement from your main text

“After correct responses were given a score of 1 and incorrect answers were given a score of 0, the first dependent variable, knowledge score, was computed for each respondent. The practice score, the second dependent variable, was computed for each respondent. 'Never = 0, rarely = 1, sometimes = 2, often = 3, and always = 4' were the response alternatives. “Your dependent variables are dichotomous, how do you categorized them? Justify it.

6.Try to cite any scientific work which supports your statement “Binary logistic regression was done to identify those candidate variables for Multiple logistic regression at a p-value of < 0.25 for not to miss the important variables.”

7.In your result part you have used only descriptive statistics. Why not Cross tab? Try to analyze your data again by using cross tab and interpret the result by considering each categories of dependent variable with each categories of independent variables. For instance: which age group of individuals have better practiced? So in order to answer questions like this you have to use cross tab rather using simple descriptive statistics.

8.Why not you haven’t seen association between the dependent and independent variable, by using Chi-square test?

9.Try to modify table 2 & 3 title, because the result is not only the binary logistic regression.

10.Interpret the cross tab result presented in the table 2 & 3, it is not interpreted.

In general your manuscript needs major revision.

6. PLOS authors have the option to publish the peer review history of their article (what does this mean?). If published, this will include your full peer review and any attached files.

Reviewer #1: **Yes: **Serebe Gebrie

Reviewer #2: No

---

## [Author Response · Author response to Decision Letter 0]

23 May 2022

To: PLOS ONE

Subject: Submitting a revised version of the manuscript

Title: Knowledge, attitude, and practice of preconception care and associated factors among obstetric care providers working in public health facilities of West Shoa Zone, Ethiopia: A cross-sectional study.

 We would like to thank the reviewers for sharing their view and experience. The comments are very important which will improve the manuscript. The point-by-point responses for each of the comment are provided in the following page:

Journal Requirements:

Authors’ Response: We have checked that our manuscript meets the PLOS ONE's style requirements

Authors’ Response: Comment accepted and we have upload our study’s minimal underlying data set as a Supporting Information files

Authors’ Response: Comment accepted and ethics statement included in the Methods section.

Additional editors’ comments:

 Overall you need to strengthen in editing, analysis and interpretation areas. You have to address the reviewer comments in bullet point wise while submitting the revised version of the paper.

Authors’ Response: Comment accepted and incorporated in the revised manuscript. We have revised the manuscript and check analysis and interpretation areas. And the manuscript is also revised for typographical errors.

Response to Reviewers’

Reviewer 1

General Comment:

In general, it is a well design study, conducted in a way scientifically and technically sound methodology. Appropriate data analysis method was employed, and figures are consistent through the document. Nevertheless, this manuscript needs editorial revision and minor changes to the abstract and discussion to make it clear and more attractive for the reader.

Detail line by line comments are provided below.

Authors’ Response: Dear reviewer, thank you so much for your constructive comments. We have included your comments and concerns in the revised manuscript.

Title: 

L112-118: the Data collection tools and procedures indicate that the study tool has five sections (demographic information, knowledge, attitude and practice, and associated factors of preconception care questions). The proceeding sentence also shows that you have assessed attitude as well. Therefore, attitude should be included in the title, keywords, and result sections.

Authors’ Response: Thanks. We included the attitude in the title, keywords, and result sections.

Since the data collection tool (questionnaire) is not provided, I couldn’t verify the actual questions that address the attitude section. Please, provide the study materials with your revised manuscript.

Authors’ Response: Comment accepted and some part of data collection tool particularly a knowledge and attitude question has been included in the revised manuscript. We have included the study questionnaire in the revised manuscript as a supporting information file

Abstract

L27-33: Please paraphrase the result section of the abstract.

Authors’ Response: Comment accepted and revisions have been made in the revised manuscript.

L27-30: has too many conjunctions “and/” and it is unclear.

Select the most relevant findings to be included in the abstract.

Authors’ Response: Comment accepted and revisions have been made in the revised manuscript.

L32: "doing screening for reproductive life plan"

oThis is a very specific task. I suggest using professional category or the department instead of pointing out a very specific task. Also, it would be much better if the result can be presented in carrier-wise like diploma, degree, midwives, physicians, and other gyn & obs care providers in the selected health facilities.

Authors’ Response: Comment accepted and revision has been made

Keywords:

Add attitude if providers’ attitude was assessed, otherwise the attitude assessment questionnaire should be removed from the tool(See comment L112-118).

Authors’ Response: Comment accepted. Attitude included in the keywords

Methods

L98: “ the first two specific objectives” - Please mention each objective here.

Authors’ Response: Comment accepted and specific objectives included

L106-107: Please, paraphrase as “ The sample was allocated to each stratum proportionally based on the number of health care providers working at the selected hospitals and health centers”.

Authors’ Response: Comment accepted. The sentence has been paraphrased in the revised manuscript

L114 - Indicates that knowledge, attitude, and practice preconception care of participants were assessed. See my comments above.

Authors’ Response: Thank you. Comment accepted!

L122 -124: Data quality assurance is beyond a questionnaire that includes data collection, data processing and data management. So, please paraphrase as

"To assure quality of the study, data collectors and supervisors were given two days training about the study materials and data collection procedures. The study tool was pre-tested among ...... "

Authors’ Response: Comment accepted and incorporated in the revised manuscript

L127: “… Cronbach's α tests which were 0.802 & 0.97 for knowledge and practice, respectively.”

Authors’ Response: Comment accepted and incorporated in the revised manuscript

L172 - “1 -to 12” correct as 1 to 12

Authors’ Response: Comment accepted and incorporated in the revised manuscript

L174 - “ 7-to 136” correct as 1 to 136

Authors’ Response: Comment accepted and incorporated in the revised manuscript

L175-176: I don't think this sentence is important. The above sentence showed that how many of them perform good practice based on your classification of good and poor.

Otherwise, you can explain as "only three participants were found to perform all the expected good practice on preconception care.... It is better focus on the meaning than the numbers.

Authors’ Response: Comment accepted and sentence removed.

L187: I think the health centers are also public. This can mislead to a comparison of public and non-public sectors.

Authors’ Response: Thank you. We have removed the word “public” to avoid confusion, because both are public institutions.

Discussion and conclusion

oPlease include strengths and limitations of this study.

Authors’ Response: Comment accepted. A study limitation has been included in the revised manuscript. 

Reviewer 2

Dear authors’ thank you for submitting this nice manuscript.

Here under some comments about your manuscript.

1. You have used special character (&) in the abstract part revise it.

Authors’ Response: Thank you! Comment accepted and special character (&) has been removed from the manuscript

2. You have used stratified sampling technique in your main text part, but also stated as simple random sampling technique in the abstract part revise it.

Authors’ Response: Comment accepted and some adjustment has been made in the revised manuscript. Dear reviewer, stratification was done for the health center and hospitals considering heterogeneity of the providers among those facilities. While simple random sampling was employed to enroll health facilities [Hospitals and Health centers] and study participants. 

3. What about the interaction effect between knowledge and practice? Try to put it clearly.

Authors’ Response: In this study, providers’ knowledge did not shows statistical significances in multivariate logistic regression. This variable was not considered for multiple logistic regressions because its p-value was 0.48 in bivariate regression.

4. What is the advantages of computing two sample size in one study? Is there any scientific study that support your ways of computing sample size? Justify it.

Authors’ Response: Calculating sample sizes for all specific objectives are crucial to compare and take the largest sample size for the study. If you have seen my cases; the calculated sample sizes were 362 and 262. If we were calculated for a single objective only, there was a probability of using small sample size, 262.

5. What is the measurement scale for your dependent variables? Means that the knowledge and practice is not directly measurable, by what criteria have you categorized it. Put the justification clearly in your document. Look the following statement from your main text

“After correct responses were given a score of 1 and incorrect answers were given a score of 0, the first dependent variable, knowledge score, was computed for each respondent. The practice score, the second dependent variable, was computed for each respondent. 'Never = 0, rarely = 1, sometimes = 2, often = 3, and always = 4' were the response alternatives. “Your dependent variables are dichotomous, how do you categorized them? Justify it.

Authors’ Response: Thank you. Comment accepted and included in the revised manuscript.

6. Try to cite any scientific work which supports your statement “Binary logistic regression was done to identify those candidate variables for Multiple logistic regression at a p-value of < 0.25 for not to miss the important variables.”

Authors’ Response: Thank you! Comment accepted and modification has been made.

7. In your result part you have used only descriptive statistics. Why not Cross tab? Try to analyze your data again by using cross tab and interpret the result by considering each categories of dependent variable with each categories of independent variables. For instance: which age group of individuals have better practiced? So in order to answer questions like this you have to use cross tab rather using simple descriptive statistics.

Authors’ Response: Dear reviewer, we did not use only descriptive statistics in the result part; we have also used logistic regression to see the association between dependent and independent variables. The age of respondents were not significantly associated with preconception care practice [p-value of ≥ 0.7] as observed in table 3. 

8. Why not you haven’t seen association between the dependent and independent variable, by using Chi-square test?

Authors’ Response: Thank you! The association between the dependent and independent variables was checked by binary logistic regression [95%CI and its AOR] was used. We believe that logistic regression is more appropriate model to answer the objective of this study. 

9. Try to modify table 2 & 3 title, because the result is not only the binary logistic regression.

Authors’ Response: Dear reviewer, cross tab result is mandatory in binary and multivariate logistic regression tables to check the value of COR manually but including such issue in the title of the table is not common and unnecessary 

10. Interpret the cross tab result presented in the table 2 & 3, it is not interpreted.

Authors’ Response: Thank you! The purposes of those tables are to show variables associated with knowledge and practice of PCC which is indicated with its p-value and 95%CI in tables and also interpreted in texts. Interpreting each cross tab result for this study might take too much spaces and meaningless in relation to its objective.

---

## [Decision Letter · Decision Letter 1]

18 Jul 2022

Knowledge, attitude, and practice of preconception care and associated factors among obstetric care providers working in public health facilities of West Shoa Zone, Ethiopia: A cross-sectional study

PONE-D-22-10177R1

Dear Dr. Negash Wakgari,

We’re pleased to inform you that your manuscript has been judged scientifically suitable for publication and will be formally accepted for publication once it meets all outstanding technical requirements. We recommend you to complete the reviewer's minor comments and *thoroughly* do the English editing.

Kind regards,

Jayanta Kumar Bora, PhD

Academic Editor

PLOS ONE

Additional Editor Comments (optional):

Reviewers' comments:

Reviewer's Responses to Questions

**Comments to the Author**

1. If the authors have adequately addressed your comments raised in a previous round of review and you feel that this manuscript is now acceptable for publication, you may indicate that here to bypass the “Comments to the Author” section, enter your conflict of interest statement in the “Confidential to Editor” section, and submit your "Accept" recommendation.

Reviewer #1: All comments have been addressed

Reviewer #2: All comments have been addressed

2. Is the manuscript technically sound, and do the data support the conclusions?

Reviewer #1: Yes

Reviewer #2: Yes

3. Has the statistical analysis been performed appropriately and rigorously? 

Reviewer #1: Yes

Reviewer #2: Yes

4. Have the authors made all data underlying the findings in their manuscript fully available?

Reviewer #1: Yes

Reviewer #2: Yes

5. Is the manuscript presented in an intelligible fashion and written in standard English?

Reviewer #1: Yes

Reviewer #2: Yes

6. Review Comments to the Author

Reviewer #1: I would like to thank the authors for their efforts to address all issues in this manuscript. They have addressed all my concerns and the manuscript is significantly improved now, except it has minor editorial errors still. I feel this manuscript can be appropriate for publication after proofreading and some editorial correction. No further peer review is required.

L16-27: Editorial correction needed.

L33 – 35: Editorial correction needed

It seems the library and internet had good knowledge than seen as positive predictors of good providers’ knowledge on preconception care.

You may revise as …

“Library availability [AOR: 1.7, 95%CI: 1.04-2.85] & and internet access [AOR: 3.4, 95%CI: 2.0-5.8] in working health facility were positive predictor of providers’ good knowledge about preconception care.”

L43 – Still it is ok, but I prefer to avoid “favorable attitude” so that not to mix the positive and negative findings in one sentence.

“ This study found that more than half of obstetric care providers’ had poor knowledge and practice of preconception care.”

Reviewer #2: Dear Author, thank you for your nice response. The manuscript does not need the further comment. No comment!

7. PLOS authors have the option to publish the peer review history of their article (what does this mean?). If published, this will include your full peer review and any attached files.

Reviewer #1: **Yes: **Serebe Gebrie

Reviewer #2: **Yes: **Lema Abate Adulo

---

## [Editor Report · Acceptance letter]

21 Jul 2022

PONE-D-22-10177R1 

Knowledge, attitude, and practice of preconception care and associated factors among obstetric care providers working in public health facilities of West Shoa Zone, Ethiopia: A cross-sectional study 

Dear Dr. Wakgari:

I'm pleased to inform you that your manuscript has been deemed suitable for publication in PLOS ONE. Congratulations! Your manuscript is now with our production department. 

Kind regards, 

on behalf of

Dr. Jayanta Kumar Bora 

Academic Editor

PLOS ONE